# Singlet Oxygen In Vivo: It Is All about Intensity

**DOI:** 10.3390/jpm12060891

**Published:** 2022-05-28

**Authors:** Steffen Hackbarth, Rayhanul Islam, Vladimír Šubr, Tomáš Etrych, Jun Fang

**Affiliations:** 1Photobiophysics, Institute of Physics, Humboldt University of Berlin, Newtonstr. 15, 12489 Berlin, Germany; 2Laboratory of Microbiology and Oncology, Faculty of Pharmaceutical Sciences, Sojo University, Kumamoto 860-0082, Japan; rayhanulislam88@gmail.com (R.I.); fangjun@ph.sojo-u.ac.jp (J.F.); 3School of Pharmacy, Queen’s University Belfast, 97 Lisburn Road, Belfast BT9 7BL, UK; 4Institute of Macromolecular Chemistry, Czech Academy of Sciences, Heyrovkého nám. 2, 16206 Prague, Czech Republic; subr@imc.cas.cz (V.Š.); etrych@imc.cas.cz (T.E.)

**Keywords:** photodynamic therapy, singlet oxygen, time-resolved phosphorescence, illumination intensity

## Abstract

The presented work addresses the influence of illumination intensity on the amount and locations of singlet oxygen generation in tumor tissue. We used time-resolved optical detection at the typical emission wavelength around 1270 nm and at 1200 nm where there is no singlet oxygen phosphorescence to determine the phosphorescence kinetics. The discussed data comprise *in vivo* measurements in tumor-laden HET-CAM and mice. The results show that illumination that is too intense is a major issue, affecting many PDT treatments and all singlet oxygen measurements *in vivo* so far. In such cases, photosensitization and oxygen consumption exceed oxygen supply, limiting singlet oxygen generation to the blood vessels and walls, while photosensitizers in the surrounding tissue will likely not participate. Being a limitation for the treatment, on one hand, on the other, this finding offers a new method for tumor diagnosis when using photosensitizers exploiting the EPR effect. In contrast to high-intensity PDT, some papers reported successful treatment with nanoparticular drugs using much lower illumination intensity. The question of whether, with such illumination, singlet oxygen is indeed generated in areas apart from vessels and walls, is addressed by numerical analysis. In addition, we discuss how to perform measurements at such low intensities.

## 1. Introduction

Singlet oxygen (^1^O_2_), the main mediator of photodynamic therapy (PDT), results from the interaction of a photosensitizer (PS) light—usually within the visible spectrum—and molecular oxygen. These three components are harmless individually, but in combination, they result in the formation of ^1^O_2_ and other reactive oxygen species [1,2,3]. Excitation of a PS results in a certain percentage in intersystem crossing to long-living triplet excited states with an energy little above that of ^1^O_2_. The PS can transfer its energy via Dexter transfer by colliding with molecular oxygen resulting in the generation of ^1^O_2_.

Due to its high reactivity, the lifetime of ^1^O_2_ in biological environments is rather short, predicted to be in the order of 40 ns [4,5] and confirmed to be shorter than 400 ns by direct observation [6,7]. Therefore, ^1^O_2_ has a maximum action radius of about 20 nm in biological environments, and consequently, in the classical sense of PDT, photosensitization affects only the cell where the PS is located. While being true in the sense of oxidation of cellular components, in tissue photosensitization in one cell also affects neighboring cells in terms of oxygen shortage [8,9,10]; the reason is simple: it is the high probability for chemical reactions of the ^1^O_2_ with cellular components, which finally causes cytotoxicity. Such reactions result in the consumption of both quenching components (resulting in cell death) and molecular oxygen. Since diffusion and solubility of oxygen in tissue [11] are much higher than in vessel walls [12], local lack of oxygen is compensated by diffusion from neighboring cells first, reducing the oxygen content there as well. Strictly spoken, partial pressure (pO_2_) is defined for gas mixtures; however, Henry’s Law relates the amount of solved oxygen to the resulting pO_2_ if there would be an atmosphere touching it, thus defining the partition coefficient. We prefer to use the pO_2_ throughout the system, as it makes the description more consistent.

While the pO_2_ in tumor tissue is in the range of 0.02 atm already because of the high metabolism of the cells, the local pO_2_ further drops with photosensitization. Depending on the amount of photosensitization this may soon result in local anoxia (pO_2_ < 0.001 atm).

It is exactly this PDT-induced anoxia that is in this paper’s center of interest. Recently, we reported the observation of this effect using the very weak yet characteristic phosphorescence of ^1^O_2_ in living mice [8]; exactly these results are finally the validation of our demand to rely on direct time-resolved spectroscopic supervision to observe ^1^O_2_. Even though this method comes along with some technical issues, the ^1^O_2_ phosphorescence correlates directly to the amount of existent ^1^O_2_; this is not the case for other related emissions, such as PS fluorescence.

As mentioned previously, in biological material, the vast majority of the generated ^1^O_2_ undergoes chemical reactions with cellular components; consequently, extravasated PS may fluoresce and phosphoresce while not generating any ^1^O_2_ due to the very low local pO_2_ caused by the photosensitization of the PS under investigation.

Former ideas to observe the impact of the local pO_2_ on the ^1^O_2_ generation pointed towards supervision of the ratio between PS fluorescence and phosphorescence intensities [13]; however, in very heterogeneous environments (as they occur *in vivo*), the reliability of steady-state observations is strongly limited. Due to the small action radius of ^1^O_2_, not every single illuminated PS necessarily contributes to the treatment impact. Biological material is highly structured and the detection volume cannot be reduced at will; therefore, signals observed *in vivo* will originate from PSs in a variety of very different environments, impossible to distinguish without at least some sort of temporal resolution. Among other questions, we aim to investigate the extent to which simple gating can cover this requirement in some cases.

Our group reported the first time-resolved ^1^O_2_ phosphorescence detection in a tumor *in vivo* through the skin of mice after systemic injection of the drug (in this case, chlorin e_6_ loaded PAMAM dendrimers) in 2016 [14]. In 2019, spatially resolved measurements of a HET-CAM model with implanted 3D-grown tumor cells [15] were published. The authors suggest a simplified method to analyze the data by just summing up certain time domains after excitation, which already gave very impressive insight into the sample under investigation. We will improve this - what we call - robust data analysis and apply it to new results gained after systemic injection of a polymer-PS conjugate into Sarcoma-bearing mice.

## 2. Materials and Methods

**The PS-copolymer:** The semitelechelic polymer precursor poly(HPMA)-NH-BOC was prepared by reversible addition–fragmentation chain transfer (RAFT) polymerization of HPMA (1.0 g, 6.98 mmol) in tert-butanol in the presence of tert-butyl *N*-[2-[(4-cyano-4-ethylsulfanylcarbothioylsulfanyl-pentanoyl)amino]ethyl] carbamate and 2,2′-azobis(4-methoxy-2,4-dimethylvaleronitrile) at 30 °C for 72 h.

The polymer precursor was isolated by precipitation in ethyl acetate, collected by filtration, washed with ethyl acetate and diethyl ether, and dried in a vacuum. The terminating trithiocarbonate group was removed as described by Perrier [16]. The BOC protecting group was removed by heating the poly(HPMA)-NH-BOC dissolved in distilled water (10 wt%/v) in a sealed ampoule at 150 °C for 1 h; then, the semitelechelic precursor poly(HPMA)-NH_2_ was obtained by lyophilization. The weight average molecular weight of the polymer precursor was *M*_w_ = 11,000 g/mol, the dispersity Đ = 1.05 and the polymer functionality of NH_2_ groups was 0.95. The polymer conjugate with pyroPheophorbide-a (pPheo) was prepared by reaction of semitelechelic poly(HPMA)-NH_2_ (400 mg) dissolved in 2.3 mL of DMSO with the pentafluorophenyl ester of pPheo (35 mg) in 2.1 mL dichloromethane.

The reaction mixture was stirred for 24 h and was then purified on a chromatography column (Sephadex LH-20, Cytiva, Uppsala, Sweden) in methanol; 350 mg of the final product (Figure 1) was obtained after precipitation in diethyl ether. The content of pPheo in the polymer conjugate was 3.0 wt%.

This polymer conjugate exploits the so-called EPR (enhanced permeability and retention) effect, although the weight average molecular weight (~11,500 g/mol) is below the commonly known mass limit of ~40 kg/mol for the preferential accumulation in tumors. PolyHPMA-pPheo forms quite stable micellar structures, resulting in a bigger hydrodynamic size of the compound, big enough for the EPR effect. Using the Zetasizer Ultra (Malvern Panalytical, Malvern, UK) with its narrow-band filter around 635 nm to cut off PS fluorescence, a hydrodynamic diameter of around 20 nm was determined. The EPR effect results from differences in the structure of tumor capillaries versus those in normal tissues, the pathological properties of tumor blood vessels (high permeability), and the limited lymphatic drainage in solid tumors [17,18].

**Phosphorescence detection**: ^1^O_2_ phosphorescence and surrounding NIR luminescence *in vivo* were recorded with a TCMPC NIR detection system (SHB Analytics, Berlin, Germany) with fiber adapter and the NIR-PMT H10330-45 (Hamamatsu, Hamamatsu, Japan). The transmission of the two optics used in this work centers around 1200 and 1270 nm with a spectral half-width of ~40 nm. For exact data on their wavelength discrimination, see [8]. When we compare measurements at the different spectral ranges in one spot, they are identical in terms of transmission multiplied by the spectral sensitivity of the detector. A laser diode Red 65X (Necsel, Cypress, USA) driven by a custom-built controller, emits pulses (FWHM 240 ns–12 kHz) at around 658 nm with an average intensity of 7 mW. The measurement duration was 30 s. The three-furcated quartz fiber (Ceram Optec, Bonn, Germany) used for these measurements consists of one excitation fiber, three fibers for simultaneous fluorescence supervision with a C10083CA fiber spectrometer (Hamamatsu, Hamamatsu, Japan), and 127 single cores (185 μm) to transport the phosphorescence signal. All fibers merge in a single sealed tip to position it directly at the place of interest, while illumination and observation volumes automatically coincide.

**Experiments in HET-CAM**: For the experiments with tumor cell-cluster bearing HET-CAM Pfitzner et al. used the same detection system, but Foslip (Biolitec, Jena, Germany) as PS/carrier and illumination at 651 nm with 10 mW average intensity. These data have been recorded using 2D scanning, thus allowing for differentiation of the less structured HET-CAM surface. Details of the experiment can be found in Ref. [15], and we thank the authors for supplying the raw data for further analysis.

**Experiments in mice**: Six-week-old ddY mice from SLC Inc., Shizuoka, Japan, were kept in standard condition with water and murine chow ad libitum. Mouse sarcoma S180 cells, grown in the peritoneal cavities of ddY mice as an ascetic form were implanted subcutaneously (2∙10^6^) in the dorsal skin of ddY mice to establish mouse S180 tumor models with a diameter of 6 to 8 mm after 7–10 days.

PolyHPMA-pPheo of 10 mg/kg (pPheo equivalent) dissolved in physiological saline (0.2 mL) was injected into the tail vein. Most results were gained 24 h after injection of polyHPMA-pPheo. Only for the comparison of signals from different origins, we use measurement data taken another 24 h after the first measurements at the now necrotic tumor. 

Apart from anesthesia with isoflurane and shaving at the areas of interest, the animals did not experience any further treatment.

## 3. Results and Discussion

Figure 2 shows the results of measurements at three different spots of the same mouse 48 h after injection of the drug and 24 h after the first measurement at one tumor (the mouse was bearing two tumors). Illumination with around 1.5 J led to necrosis in the tumor after 24 h, covering the central part and about half the volume of the tumor.

While the PS fluorescence was the brightest in the necrotic tumor, there was no ^1^O_2_ phosphorescence detectable (Figure 2 bottom). The kinetics determined at 1270 nm, the characteristic wavelength of ^1^O_2_ phosphorescence, is identical to that determined at 1200 nm, where we only record the PS phosphorescence. The higher intensity at 1200 nm corresponds to the wavelength dependence of the PS phosphorescence (maximum at 932 nm [19]).

The kinetics determined at the normal tumor (Figure 2 middle) also show an intensive slow decaying signal, indicating anoxia. The signal comprises mainly PS phosphorescence of the extravasated drug [8]. In addition, in the first few µs after excitation (grey area) one can see that a weak ^1^O_2_ phosphorescence is also visible here; these signals from the tumor are very similar to those published previously [20].

In the healthy tissue, a very clear ^1^O_2_ signal and nearly no PS phosphorescence (Figure 2 top) are visible. The same is true for PS fluorescence (Figure 2 right). The kinetics can be described with the typical double exponential model using rise and decay times of 1.0 ± 0.2 µs and 7.5 ± 1.0 µs, and are thus closer to typical values for ^1^O_2_ luminescence in blood [21].

As an outcome, we can clearly distinguish healthy tissue from tumors and necrotic tumors. In both the latter cases, there is extravasated PS, resulting in very low pO_2_ upon illumination and, thus, mainly PS phosphorescence instead of ^1^O_2_ luminescence.

To answer the question of whether our assignment of the signal components is correct, a measurement with comparable detection but a higher selectivity is required, which allows us to distinguish between blood vessels, tumor tissue, and the rest; indeed, Pfitzner et al. [15] did such a measurement. We wanted to test whether our simple model with mainly three signal components (^1^O_2_ phosphorescence from the blood, PS phosphorescence from extravasated drug, and the ever-present short-time artifact in such measurements) would stand the test.

Our “robust data analysis” works as follows: Step 1 is a tail-fit starting at 15 µs and comprising two exponentially decaying signal components. The two decay times then represent the falling flank of the ^1^O_2_ phosphorescence in the blood and the long decaying signal of extravasated PS. Of course, any ^1^O_2_ phosphorescence originating outside the vessels would decay with the same long decay time, but as shown later, at the excitation intensities as we apply them so far, the vast majority of the signal is PS phosphorescence. In step 2, we subtract both the determined slow decaying signal component (monoexponential) and the background from the signal and integrate the resulting signal, omitting the first 1 µs. The result corresponds to the red area marked with A in Figure 3a. In step 3, we integrate the background-corrected signal, starting at a time when the ^1^O_2_ phosphorescence coming from the blood decayed. We chose 40 µs as a start point, hence being more than 5 times the PS triplet decay time in blood. The result corresponds to the blue area marked with B in Figure 3a.

Applying this procedure to the 2D phosphorescence data of the HET-CAM model carrying the implanted 3D tumor gives impressively clear results (Figure 3b). While ^1^O_2_ phosphorescence originates nearly exclusively from the blood vessels, the PS phosphorescence clearly indicates extravasated PS in areas suffering from anoxia, which coincide with the tumor; however, there is one small area outside the tumor, where both signals are present. The most likely explanation is a slight rupture of a blood vessel causing a certain leakage and thus PS extravasation. The fluorescence image (Figure 3d) supports this assumption and was taken right after the image shown in Figure 3c. Exactly at the point of interest, the blood vessel is fluorescing less than expected. After measurement, when all illuminated vessels face some rupture (Figure 3e), they all are darker in the fluorescence image (Figure 3f).

We may thus conclude that we find evidence to suggest that PS phosphorescence in tissue is nearly exclusively originating from extravasated PSs under local anoxia caused by their own photosensitizing activity. If this conclusion holds for tumors in mice, we have found a promising new method for tumor diagnostics.

We made a first “scanning” experiment in a sarcoma mouse model, with “scanning” set in quotations marks as it was performed by placing the fiber tip manually at 11 spots along a line crossing the tumor at the back of the mouse. The black dots in Figure 4 indicate the detection spots in the mouse image/fluorescence overlay in the background. They also represent the abscissa for the results of classical fitting of the determined data and the robust data analysis. The results for the quantitative evaluation of the ^1^O_2_ phosphorescence are nearly identical for both methods. Again, despite the fact that much more PS is located in the tumor, the ^1^O_2_ generation of these molecules is low; however, their phosphorescence allows reliable identification of tumors. Even better, for such a purpose there is no real need for time-resolution, gated detection is sufficient. All data in Figure 4 are corrected for thermal background, which for a typical measurement under the given conditions was Poisson-distributed noise with an average of 14 ± 0.5 counts per channel.

On the one hand, we found that EPR-based PS might potentially act as sensor molecules for a new kind of tumor diagnosis; however, on the other hand, one particular question arises: is it possible to detect ^1^O_2_ phosphorescence from the tumor tissue at all or will ^1^O_2_ always be limited to the blood vessels and their walls?

Most illumination intensities suggested for commercial PDT drugs (e.g., 100 mW/cm² for Foscan, according to the leaflet) are much higher than those we used for our measurements. One might speculate that most of such treatments cause destruction of the tumor vessels, which would certainly also have an impact on tumor development. This is, of course, a valid treatment strategy, but it should be intended.

It is worth noting that the group of Prof. Maeda reported a complete cure of a DBMA-induced rat breast cancer model *in vivo* using HPMA polymers loaded with Zn Protoporphyrin [22]. One injection and three illumination sessions with broadband light were able to eradicate the tumors. One interesting detail in this study is the very weak absorption of ZnPP in the wavelength region, which can penetrate the skin. Furthermore, the illumination with broad spectral range light results in only a very small part of the total light intensity actually exciting the PS. Correcting for the PS absorption, the effective excitation intensity was about 25 times lower than during our measurements. Of course, the illumination time was long enough to deliver a sufficient light dose; nevertheless, it would be interesting if ^1^O_2_ phosphorescence detection is possible at such intensities. In that context, two more questions arise: (1) Can we arrange a measurement with such low excitation intensity? (2) Will it be possible to detect ^1^O_2_ phosphorescence originating really from inside the tumor tissue? Fortunately, there is an answer to both questions:
(1)In this work, we used the standard version of the H10330-45 PMT. There is a modified version available (SHB Analytics GmbH) with about three times higher etendue, while in the meantime, the quantum efficiency of the NIR-PMTs increased. Altogether, this allows a factor of 5 improvement in detection sensitivity; however, the solution for the remaining problem is the pulsed time-resolved detection. In the end, our detection consists of 360,000 excitation pulses. The SNR of the measurement remains unchanged, independent of the time duration between two pulses, which allows a free scaling of intensity, but at the cost of a long measurement duration.(2)Based on the Krogh tumor model that we justified in [8], we calculated signal kinetics that corresponds to differnt illumination intensities. The model comprises a blood vessel of 50 µm diameter, surrounded by 15 µm vessel wall and a further 45 µm of tumor tissue. The required parameters for oxygen diffusion and solubility, as well as the corresponding references, are summarized in Table 1. To keep the model simple, we fix pO_2_ = 0.12 atm in the vessel up to a radius of 20 μm, based on values reported in [23]; therefore, it has little influence that we approximate the diffusion coefficient of blood with the one of water. 

Calculating the oxygen solubility of blood is difficult. Chistmas et. al. [28] reported the oxygen solubility in plasma to be comparable to that in water, but in reality, erythrocytes increase this value; however, this parameter only influences the simulation at the blood/vessel wall interface. It is certain that the value is bigger than that of the vessel wall. Regardless, lower oxygen diffusion and solubility in the wall are the most limiting factors for oxygen transport. For this simulation, we chose the solubility in blood to be comparable to that in tissue; this estimation comes with a big error margin, but the variation of this value by 50% in each direction had no recognizable impact on the results. Furthermore, we estimate the PS concentration in the tissue as 20 µM, the absorption cross-section at the excitation wavelength as 1.5∙10^−16^ cm^2^, and the ^1^O_2_ quantum yield in cells as 0.25 [8], so at about 50% of that in solution.

To adapt the simulation to other PSs, use a comparable product of photon flux density, absorption cross-section, concentration, and ^1^O_2_ quantum yield. As an example, a PS with half the singlet oxygen quantum yield gives the same results at twice the illumination intensity.

Finally, we assume 80% of the generated ^1^O_2_ to react with cellular components [7] and the drug concentration in blood to be at 10% of that in the tissue. This is a reasonable assumption because the drug clears from the blood but not from the tissue, resulting in this or a similar ratio at a certain time after injection. Figure 5 shows the calculated signal kinetics for excitation with 66 mW/cm² (the intensity used in this work), 6.6 mW/cm², and 0.66 mW/cm².

Assuming that all signal at 1270 nm is either PS phosphorescence or ^1^O_2_ phosphorescence, the local pO_2_ determines the corresponding triplet decaytime τ_T_ of the PS and thus the signal kinetics at each spot:$$I\left(t\right)=\gamma \cdot \frac{\tau_{T,0}-\tau_{T}}{\tau_{T,0}}\cdot \frac{\tau_{\Delta }}{\tau_{T}-\tau_{\Delta }}\cdot \left[\exp\left(-\frac{t}{\tau_{T}}\right)-\exp\left(-\frac{t}{\tau_{\Delta }}\right)\right]+\frac{\tau_{T}}{\tau_{T,0}}\cdot \frac{\rho }{\tau_{T}}\cdot \exp\left(-\frac{t}{\tau_{T}}\right)+BG$$*γ* and *ρ* are constants, but proportional to the radiative rate constants of ^1^O_2_ and PS phosphorescence. Parameters such as setup geometry and sensitivity also affect these values, but since we only want to analyze the signal composition here, we only need the ratio of *ρ*/*γ*, which we may estimate as 6 ± 1 from measurements in various solutions and suspensions under varying pO_2_. The index 0 indicates a triplet decay time in the absence of oxygen. We set *τ*_Δ_ = 0.4 µs in the tissue, which is the values we found for this PS in cells in vitro [7]. For signal components originating in blood, we assume them to follow the kinetics found in blood before (1.5 and 7 µs) [21].

For a given illumination intensity, the simulation calculates the pO_2_ in concentric rings with the vessel being in the center. For each of these rings, the corresponding kinetics contribute to the overall signal, taking the volume of each ring as well as its PS concentration into account. Figure 5 shows the sum signal, separated into the share from different origins.

At the intensities that used so far, the ^1^O_2_ signal is originating nearly exclusively from the vessel wall, which explains their severe damage after illumination (Figure 3). According to the model, there is no ^1^O_2_ generated in the tumor tissue.

At lower intensities, similar to those applied by the Maeda group in the aforementioned study, the model shows ^1^O_2_ generation in the tissue, which may be the explanation of why these experiments were successful in curing the tumors. Further reduction of the illumination intensity finally results in a superior amount of the signal originating from the tumor tissue.

## 4. Conclusions

We could demonstrate with experiments in tumor-laden HET-CAM and mice, that photosensitization following high illumination intensities results in anoxia in all regions with extravasated PS, hence in a best-case scenario, the tumor tissue. In such regions, oxygen is exclusively available in the bloodstream and the vessel wall. High intensities in this context means just a few mW/cm², depending on the extinction, concentration and singlet oxygen quantum yield of the PS in the tissue. The bottleneck for photosensitization is the oxygen supply. Several consequences arise from this finding. On one hand, induced anoxia in combination with an EPR-based PS offers a promising new diagnostic tool for tumor detection with high contrast. On the other hand, one may assume that most practiced PDT treatments affect the blood vessels of the tumor only.

While at such high illumination intensities ^1^O_2_ phosphorescence detection and analysis is now possible, the more interesting cases are those with lower illumination that avoid anoxia; however, we could show that such measurements are useful and technically possible. We will follow up on these ideas in the near future.

## Figures and Tables

**Figure 1 jpm-12-00891-f001:**
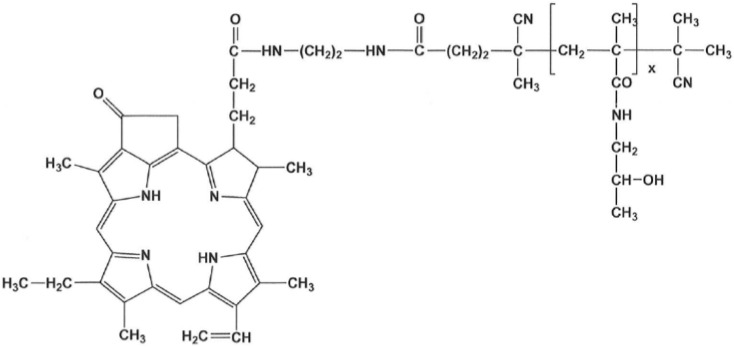
The semitelechelic polyHPMA-pPheo conjugate was used for the experiments in mice.

**Figure 2 jpm-12-00891-f002:**
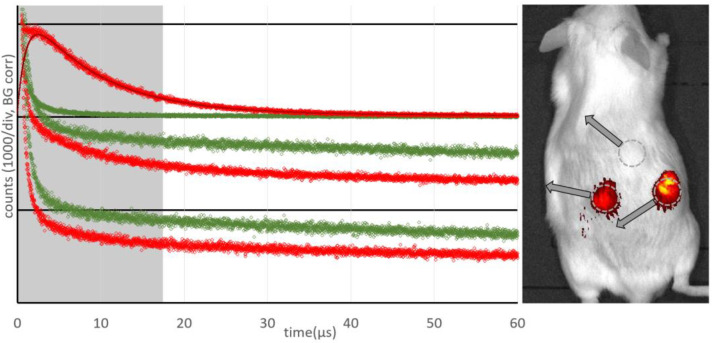
Phosphorescence kinetics *in vivo* recorded for healthy tissue including a typical fit (**top**), at a tumor (**middle**) and at a necrotic tumor (**bottom**) for wavelengths around 1200 nm (green) and 1270 nm (red). Detection spots are indicated in the fluorescence image overlay of the mouse with a cutoff at 30% of maximum intensity (right).

**Figure 3 jpm-12-00891-f003:**
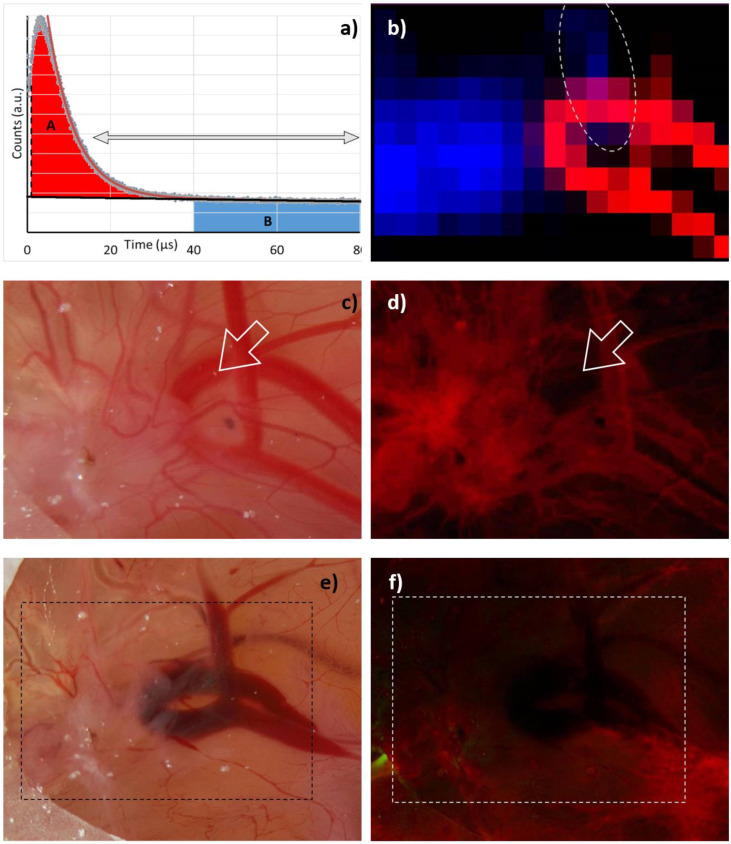
(**a**) Robust data analysis based on a double exponential decay fit (red curve) within the time range indicated by the arrows (15–80 µs). The longer decay time from this fit is considered to be the lifetime of PS phosphorescence in tissue (black). The relative ^1^O_2_ intensity is determined from the difference between signal and PS phosphorescence (light red marked area - A) omitting the first 1 µs. Relative PS phosphorescence intensity is determined from the blue marked area (B) from 40 to 80 µs. (**b**) 2D plot of A (red) and B (blue) as determined at gritted positions across the HET-CAM area. Values below 10% of the corresponding maximum value are black. Photograph and fluorescence image of the investigated area before (**c**,**d**) and after (**e**,**f**) measurement.

**Figure 4 jpm-12-00891-f004:**
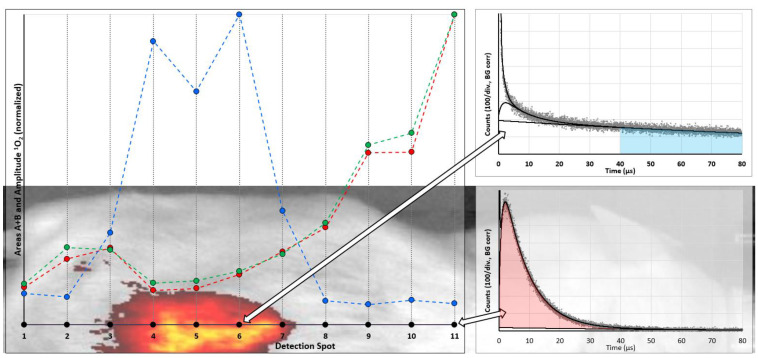
Results of measurements at 11 spots (black dots) along a line across the mouse skin, crossing the tumor. The overlay with the fluorescence image of the mouse indicates the exact locations. PS phosphorescence intensity was determined by summing up the signal in the range of 40–80 µs after laser pulse, corrected for background (blue). Relative ^1^O_2_ intensity is shown for two ways to analyze the data: the amplitude of the ^1^O_2_ component determined by fitting of the kinetics (green) and robust data analysis as explained in the text (red). For detection spots 6 and 11 the determined phosphorescence kinetics are shown on the right (indicated by the arrows), where red and blue areas are again the graphical representation of the values determined by the robust data analysis.

**Figure 5 jpm-12-00891-f005:**
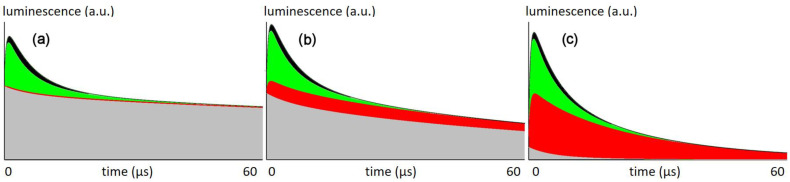
Simulated data for the phosphorescence kinetics in tumors for illumination intensities of (**a**) 66 mW/cm², (**b**) 6.6 mW/cm² and (**c**) 0.66 mW/cm² based on the simulation parameters mentioned in the text and Table 1. Graphs show the calculated luminescence signals for the first 60 µs after excitation pulse, the arbitrary units used here will scale with illumination intensity. Colors indicate signal contributions according to origin and type: ^1^O_2_ in blood (black), ^1^O_2_ in vessel wall (green), ^1^O_2_ in surrounding tissue (red) and total PS phosphorescence (grey).

**Table 1 jpm-12-00891-t001:** Parameters as used for the numerical simulation of the Krogh model.

	Blood	Vessel Wall	Tumor Tissue
O_2_ solubility (µmol/cm³/atm)	2 (see text)	1 [12]	2 [11]
O^2^ diff. coefficient (10^−6^ cm²/s)	20 (water)	2 [12,24]	24 [25]
O^2^ consumption (nmol/cm³/s)	0	11 [12,26]	20 [27]

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
