# Peer review of "Singlet Oxygen In Vivo: It Is All about Intensity"

_jpm, 2022, doi:10.3390/jpm12060891_

Round 1

Reviewer 1 Report

The authors present a methodology to measure singlet oxygen in tumors for PDT and show the relationship with the light intensity.

The manuscript is well explained and easy to follow, and data are shown to support evidence.

The authors just need to address a couple of questions:

  • Do you think the influence of light intensity is independent from the ability of the PS to generate singlet oxygen? What is the Singlet Oxygen Quantum yield of the PS? Please explain.
  • Point 3, change Diskussion by Discussion

Author Response

First of all we want to thank both reviewers for the very fast work they have done.

The authors present a methodology to measure singlet oxygen in tumors for PDT and show the relationship with the light intensity. The manuscript is well explained and easy to follow, and data are shown to support evidence.

Thanks for this statement.

The authors just need to address a couple of questions:

  • Do you think the influence of light intensity is independent from the ability of the PS to generate singlet oxygen? What is the Singlet Oxygen Quantum yield of the PS? Please explain.

A good approximation is that the effect scales directly with the singlet oxygen quantum yield. Unfortunately, so far, no value for FD in biological samples has ever been published (to my knowledge). I have a student working on exactly this topic to give a first answer. So far we need to work with a reasonable guess, which we made in our paper in 2019, based on values determined in Tween. I put the reference indicator at the correct place this time, so one knows why we assume 0.25. (green text) We also included a more illustrative explanation how to transfer our results to other photosensitizers. (red text)

  • Point 3, change Diskussion by Discussion

Thanks, that is a typical German typo. Corrected….and some others as well.

Reviewer 2 Report

The definite value of the MS „Hackbarth et al.,: Singlet oxygen in vivo – it is all about intensity” (jpm-1714468) is the carefully designed and conducted time resolved in vivo detection of phosphorescence of singlet oxygen at its specific emission wavelength. For this respect the MS has great of importance in PDT in medicine and will be interesting for the scientific community.

I have to state that the data look valuable, however the readability of the MS is not easy. Unfortunately, I can’t provide row-by-row evaluation because of the large numbers of mistakes. I will give here typical examples only.

1. Main scientific concerns:

- Is it possible to provide the pre-illumination background signal of the kinetic measurements? Without this it is hard to figure out the true background for further analysis.

- Also, it would be good to see the fitted lines presented on the signals and the fitted parameters (the latter arranged, e.g., in separate tables).

- Please, provide abscissas and ordinates of Figs. 5.

- Henrys Law is true not only in atmospheric conditions. Actually „Henry’s law”. (L: 48)

- There are large numbers of non-proper expressions – examples are here:

“around it” and “close to it” do not sound “scientific”;

„…diffusion and solubility of oxygen in tissue is much better than… ”- „much better„ also not scientific;

“small structures” ?

“We want to improve it…” – do the authors want or did it?

„It was taken right after image (Fig. 3c).” ?

- “Henrys Law” is true not only in atmospheric conditions. Actually „Henry’s law”.

2. Grammar, readability

- There are large number of mistypes (typesetting, inconsistent using of decimal markers and subscripts of chemical signs).

- Please, check reference formatting to the journal style.

- Extensive English reediting is necessary.

I recommend major revision and reediting.

Author Response

First of all we want to thank both reviewers for the very fast work they have done.

The definite value of the MS „Hackbarth et al.,: Singlet oxygen in vivo – it is all about intensity” (jpm-1714468) is the carefully designed and conducted time resolved in vivo detection of phosphorescence of singlet oxygen at its specific emission wavelength. For this respect the MS has great of importance in PDT in medicine and will be interesting for the scientific community.

Thanks for this positive statement.

I have to state that the data look valuable, however the readability of the MS is not easy. Unfortunately, I can’t provide row-by-row evaluation because of the large numbers of mistakes. I will give here typical examples only.

We are aware, that we are non-natives. That is why we always give our texts to qualified persons for language workover. We again checked all the text for quality. We made several small changes but not highlighted, as they do not change the scientific content. That includes a number of upper-lower case or space. We hope that the reviewer finds it more convenient to read now.

  1. Main scientific concerns:

- Is it possible to provide the pre-illumination background signal of the kinetic measurements? Without this it is hard to figure out the true background for further analysis.

We agree, it is a real improvement to include such data in the text. However, a figure seems not necessary, since the following addition should give all the required info:

„All data in Fig 4. are corrected for thermal background, which for a typical measurement under the given conditions was Poisson-distributed noise with an average of 14 +/- 0.5 counts per channel.“

Some more info for you to explain, why we go for just the text:

The uploaded screenshot shows background noise and its histogram from 3900 channels (some electronic artifact outside the measurement window are taken out) of a typical pre-illumination background that we took before the presented experiments. Fitting the background results in a red Chi2 of 0.988. The histogram (blue dots) matches the theoretical Poisson distribution (red dots).

(see attachment)

- Also, it would be good to see the fitted lines presented on the signals and the fitted parameters (the latter arranged, e.g., in separate tables).

In Fig. 4 two of the fits are show exemplarily. Fit models are not the subject of this manuscript - we explained them in former papers, to which we refer, more would not give an extra value and rather distract from the main message.
With the background signal explained (which was a good hint), everyone can see the high contrast of this method. The message of this work is the contrast and the fact that just gated detection can do the trick. Raw data will give no added value. Also, according to the rules of good science, they are always available in our data base, for everyone, for at least ten years.

In fig. 2 we just want to show that the differences between three different spots can be seen even by the naked eye. However, we included the typical fit for those data, where fitting makes sense and where we named parameters.
The values are 1.05 µs and 8.5 µs for exactly this fit. The values mentioned in the text are an average of many measurements.

- Please, provide abscissas and ordinates of Figs. 5.

They were described in the caption to save space, but we agree, it is better if one can see it immediately by looking at the image.

- Henrys Law is true not only in atmospheric conditions. Actually „Henry’s law”. (L: 48)

Henrys Law describes the concentration in relation to a partial pressure in an ideal gas and a partial pressure is defined in gasses. Non-atmospheric conditions is a theoretical idea that does not exist in reality.
Of course, there will be oxygen solved also if the solvent is in a closed container, but the solved amount still corresponds to the partial pressure that would exist, if the solvent had contact to an atmosphere. In fact, there will be fluctuations and if the pressure is reduced (the container getting flexible), then an atmosphere is formed from oxygen leaving the solvent.
However, to cover also the theoretical case, the text is changed to:

„However, Henrys Law relates the amount of solved oxygen to the resulting pO2 if there would be an atmosphere touching it, thus defining the partition coefficient.

- There are large numbers of non-proper expressions – examples are here:
“around it” and “close to it” do not sound “scientific”;
„…diffusion and solubility of oxygen in tissue is much better than… ”- „much better„ also not scientific;
“small structures” ?

Wherever we found such expressions, we corrected them. Bigger modifications are marked.

“We want to improve it…” – do the authors want or did it?

These words were used at the end of the introduction, where the reader is prepared for what will be reported soon, hence the classical way to write a scientific paper. At this place, it is still a plan that will be realized later.

However, we replace „want“ by „will“, so it is no longer a plan, but an announcement.

„It was taken right after image (Fig. 3c).” ?

Modified.

  1. Grammar, readability

- There are large number of mistypes (typesetting, inconsistent using of decimal markers and subscripts of chemical signs).

- Please, check reference formatting to the journal style.
- Extensive English reediting is necessary.
I recommend major revision and reediting.

References received a workover and actually one correction.

Round 2

Reviewer 2 Report

First of all, I can emphasize again that the scientific value and the message of the MS Hackbarth et al.,: Singlet oxygen in vivo – it is all about intensity” (jpm-1714468) is important for the scientific community and worth publishing.       

After the first revision the readability is better now, however, I still see few trivial mistakes and scientific questions should, probably, be addressed.

1. To be precise, Henry’s law describes that “the amount of dissolved gas in a liquid is proportional to its partial pressure above the liquid”. This is nothing to do with the atmospheric pressure (as suggested in P.3,L.49). Just note that pO2 is different even in the atmosphere and the alveolar system. Otherwise, I agree with using this parameter “throughout all the system” (citation from the text).

By the way, the correct writing is “Henry’s law” and not “Henrys law”. Also, “atm” is forbidden in SI (the journal can decide that it fits to its style or not).

2. Regarding the question of the preillumination background I disagree with the authors’ answer. In this presentation and interpretation the decay curves of “Fig. 2” don’t look bi-exponential. I don’t see that the slow decay components (several 10-s of ms-s) are real decay components or the slow drift of the background. Nevertheless, I trust the experience of the authors, and on Figs. 3 and 4 we get some impression about it.

 3. Still there are trivial mistakes.

- Please, check decimal signs (dot or comma?) (e.g. “molecular weight (~11,5 kDa)”; “around 1,5 J led to”; “decay times of 1,0 +/- 0,2 μ s and 7,5 +/- 1,0 μ s”).

By the way, according to SI „molar mass” should be written instead of molecular weight”.

 - There are still lot of “space” mistakes (also in the list of references). The editors will decide how to handle them.

Personal remark: Authors answered that the text has been checked by an expert. If so, how is it possible that “qualified person” did not give a warning of large numbers of trivial mistakes?

 In summary, I agree with publishing this work and suggest a final careful correction again, accordingly.

Author Response

General Commends:

First of all, I can emphasize again that the scientific value and the message of the MS „Hackbarth et al.,: Singlet oxygen in vivo – it is all about intensity” (jpm-1714468) is important for the scientific community and worth publishing.       

After the first revision the readability is better now, however, I still see few trivial mistakes and scientific questions should, probably, be addressed.

 In summary, I agree with publishing this work and suggest a final careful correction again, accordingly.

Thanks for this suggestion and once again thanks for the fast review. We did our best to find as many of the remaining mistakes as possible… no guarantee for completeness, though.

Scientific Qustions:

1.To be precise, Henry’s law describes that “the amount of dissolved gas in a liquid is proportional to its partial pressure above the liquid”. This is nothing to do with the atmospheric pressure (as suggested in P.3,L.49). Just note that pO2 is different even in the atmosphere and the alveolar system. Otherwise, I agree with using this parameter “throughout all the system” (citation from the text).

The reviewer mixes different things here. While Henry’s Law exactly relates to the partial pressure in the athmospere in contact in a quasi static system (this is what it is defined for), the alveolar system is highly dynamic. The system is trying to reach this point, but the permeability of the layer inbetween is limited and so the blood leaves the scene, before the final parameter is reached. Due to the diffusion “resistance” of the alveolar membrane the concentration of oxygen in the blood approaches in an exponetial manner the partion it would have in a static system. By the way, this is exactly what our paper is all about, limitation by permeability.
Nevertheless, Henry’s Law applies, the partition coefficient is a material constant and not dependent on the surroundings. As a consequence, behind a diffusion limiter the partial pressure is lower. One can imagine a local athmospheric contact at each point of the system that corresponds to it.

By the way, the correct writing is “Henry’s law” and not “Henrys law”. Also, “atm” is forbidden in SI (the journal can decide that it fits to its style or not).

That is a typical “German” mistake, thanks for the hint.

Concerning the atm, yes it is not SI, but due to practicability, it is commonly used for describing solubility of oxygen. Examples:
Wei Xing, ... Jiujun Zhang, in Rotating Electrode Methods and Oxygen Reduction Electrocatalysts, 2014
Pauline M. Doran, in Bioprocess Engineering Principles (Second Edition), 2013

Some even refer to bar or torr, since the usual reader of such papers is using such scales:

DOI:10.1016/J.GCA.2010.06.034Corpus ID: 38137595 Prediction of oxygen solubility in pure water and brines up to high temperatures and pressures M. Geng, Zhenhao Duan  Published 1 October 2010    Chemistry   Geochimica et Cosmochimica Acta

In the internet, these scales are everpresent:
https://www.google.com/search?source=univ&tbm=isch&q=solubility+oxygen+water

Still, if the editor insists, we may change the scale, even though I would not recommend it.

  1. Regarding the question of the preillumination background I disagree with the authors’ answer. In this presentation and interpretation the decay curves of “Fig. 2” don’t look bi-exponential. I don’t see that the slow decay components (several 10-s of ms-s) are real decay components or the slow drift of the background. Nevertheless, I trust the experience of the authors, and on Figs. 3 and 4 we get some impression about it.

Background drift is an effect of analogue detectors; here we use a counting system. Dark counts of an eventbased detector is dependent on the temperature only and this is kept constant at -70°C. The reviewer should believe us, if we say there is NO drift, at least not on a scale shorter than years (there is some drift due to detector ageing). I invite the reviewer to come to our labs and look at our technology. Also, if you want to have such a system for your labs, contact our spin-off at SHBA.de

Background counts, other than dark counts are black body emission of the sample, but the mice have a very similar temperature, and background light, which we learned to avoid.

Formal concerns:

- Please, check decimal signs (dot or comma?) (e.g. “molecular weight (~11,5 kDa)”; “around 1,5 J led to”; “decay times of 1,0 +/- 0,2 μ s and 7,5 +/- 1,0 μ s”).

By the way, according to SI „molar mass” should be written instead of “molecular weight”.

 - There are still lot of “space” mistakes (also in the list of references). The editors will decide how to handle them.

As said above, we did our best. If the editor thinks, there are still too many typos, even thought I don’t think, it will affect the readability of the paper, I am willing to pay one hour ($200,-) for someone doing a final formal workover to avoid this “trial and error”. Decimal signs should be dot, but it is different in several countries and therefore, even the Word autocorrection partly acts against us.
Polymers are defined by weight average molecular weight Mw, number average molecular weight Mn and Dispersity D, where D is defined as Mw/Mn.

Between reference numbers and in chemical formulae no space after comma is intentionally. Also, no space before and after “/” as explained at https://www.grammarly.com.

Personal remark:

Authors answered that the text has been checked by an expert. If so, how is it possible that “qualified person” did not give a warning of large numbers of trivial mistakes?

Since this remark violates the standards of a good scientific discussion, I will not commend on it.